# Free Association Database for a 62-Word Dataset Including Emotion and Colour Terms in English, Estonian, French, German, Italian, Lithuanian, and Spanish: Data from 14 Countries

Journal of **open** psychology data

**DATA PAPER**

]u[ ubiquity press

**DOMICELE JONAUSKAITE**

**DÉBORAH EPICOCO**

**MALIHA BOUAYED MEZIANE**

**BRITT BURTON**

**VIOLETA CORONA**

**EDUARDO FONSECA-PEDRERO**

**CONSUELO GONZÁLEZ-DÁVILA**

**JELENA HAVELKA**

**ERIC LAURENT**

**BIGNA LENGGENHAGER**

**TOBIAS LOETSCHER**

**STEPHANIE LOPEZ CASTIÑEIRA**

**LEILA MANNI**

**PHILIP MEFOH**

**DANIEL OBERFELD**

**MERLE OGUZ**

**RABIA YAĞMUR ÖZDURAN**

**CORINNA PERCHTOLD-STEFAN**

**PATRICIA QUANT**

**MICHAEL QUIBLIER**

**MAJA ROCH**

**SUDE SARAYKÖYLÜ**

**GIULIA F. M. SPAGNULO**

**MAËL THEUBET**

**CECILIA TOSCANELLI**

**MARI UUSKÜLA**

**CHRISTINE MOHR**

*Author affiliations can be found in the back matter of this article

**CORRESPONDING AUTHOR:**
**Domicele Jonauskaite**

Institute of Psychology, University of Lausanne, Lausanne, Switzerland

domicele.jonauskaite@unil.ch

**KEYWORDS:**
Semantic networks; Cross-linguistic semantics; Cross-cultural psychology; Multilingual database; Psycholinguistics

**TO CITE THIS ARTICLE:**
Jonauskaite, D., Epicoco, D., Meziane, M. B., Burton, B., Corona, V., Fonseca-Pedrero, E., González-Dávila, C., Havelka, J., Laurent, E., Lenggenhager, B., Loetscher, T., Castiñeira, S. L., Manni, L., Mefoh, P., Oberfeld, D., Oguz, M., Özduran, R. Y., Perchtold-Stefan, C., Quant, P., Quiblier, M., Roch, M., Sarayköylü, S., Spagnulo, G. F. M., Theubet, M., Toscanelli, C., Uusküla, M., & Mohr, C. (2025). Free Association Database for a 62-Word Dataset Including Emotion and Colour Terms in English, Estonian, French, German, Italian, Lithuanian, and Spanish: Data from 14 Countries. *Journal of Open Psychology Data*, 13: 4, pp. 1–17. DOI: https://doi.org/10.5334/jopd.140

## ABSTRACT

This article presents a free association database containing responses to 62 stimulus words (including colour terms, emotion words, and common nouns) across seven languages: English, Estonian, French, German, Italian, Lithuanian, and Spanish. Data were collected online from 1,439 participants (mean age 31.47 years) across 14 countries, yielding 223,786 responses. All data were cleaned, normalised, and organised for analysis, with both raw and processed datasets available on OSF. This cross-linguistic resource enables research on semantic networks, psycholinguistics, translation studies, and cross-cultural comparisons, providing insights into how meaning is constructed within and across different languages and cultures.

# 1. BACKGROUND

The open data movement has gained momentum in psychology over the past decade, responding to the field's ongoing concerns about reproducibility (Munafò et al., 2017; L. D. Nelson et al., 2018). In this regard, research depends on robust, accessible datasets that enable replication, facilitate study design, and allow for novel methodologies and analyses to be run in new populations. Making psychological datasets truly Findable, Accessible, Interoperable, and Reusable (FAIR; Wilkinson et al., 2016) is crucial for advancing research and addressing reproducibility concerns (Munafò et al., 2017; L. D. Nelson et al., 2018). Despite the fundamental scientific value of shared data (Meyer, 2018), many datasets remain either completely unpublished or isolated in lab-specific repositories, where the lack of documentation and standardisation limit their utility. This data article contributes a new, cross-linguistic resource designed with FAIR principles at core.

Over recent years, there have already been notable endeavours to provide researchers with extensive datasets, focused on language and cognition (e.g., Bradley & Lang, 1999; Brysbaert et al., 2014; Brysbaert & New, 2009; De Deyne et al., 2019; Duyck et al., 2004; Keuleers et al., 2012; Kuperman et al., 2012; Lenci et al., 2013; Lynott et al., 2020; Monnier & Syssau, 2017; D. L. Nelson et al., 2004; Stadler et al., 1999; Sutton & Altarriba, 2016; Warriner et al., 2013). These datasets are important because well-designed studies require well-controlled linguistic material, which is effortful to collect. Word sets are commonly controlled for word frequency, length, imageability, valence, arousal, age of acquisition, and knowledge of other languages (e.g., Brysbaert et al., 2016, 2018; Keuleers et al., 2015; Kuperman et al., 2014; Warriner & Kuperman, 2015). When a subsequent study is conducted in another language, these norms quickly change, requiring new norms and likely also new word sets (e.g., Willemin et al., 2016).

Challenges are also high for semantic association studies, in which the structure of knowledge and abstract representation are commonly tested by having people spontaneously report what comes to their mind when being presented with a trigger stimulus. Already Sigmund Freud (1913) used free associations to understand his patients' minds by having them freely express emerging thoughts (also see, Lothane, 2018). In clinical settings, free associations remain in regular use (for an overview, see Novac & Blinder, 2021). They are also popular in experimental fields, such as creativity (Merseal et al., 2025), language learning (Citraro et al., 2023; Mak & Twitchell, 2020), cognition (De Deyne et al., 2013; Muraki & Pexman, 2024), and computational modelling of semantic networks (Steyvers & Tenenbaum, 2005). Indeed, word associations are thought to reflect the underlying structure of human semantic memory and provide a window into how meaning is constructed within and across languages and cultures (D. L. Nelson et al., 2000).

Large-scale word association datasets have been published for some languages, such as English (De Deyne et al., 2019), Dutch (De Deyne & Storms, 2008), or Spanish (Cabana et al., 2023). However, we know of no comprehensive dataset on semantic associations for the same word set in different languages. Due to our interest in the cross-cultural links between colour and emotion (e.g., Jonauskaite et al., 2019, 2020, 2024; Jonauskaite & Mohr, 2025; Ram et al., 2020; Uusküla et al., 2023; Weijs et al., 2023), we had collected free association data for colour terms, emotion terms, and various 'filler' words in English, Estonian, French, German, Italian, Lithuanian, and Spanish.

Here, we present this complete dataset. It contains 223,786 free association responses from 1,439 participants across 14 countries who provided associations to 62 stimulus words in seven languages. The stimulus words comprise 11 basic colour terms (e.g., *red*, *green*, *blue*), 5 non-basic colour terms (e.g., *turquoise*, *beige, violet*), 20 emotion terms (e.g., *love, joy, sadness, anger*), and 26 common nouns, the latter including eight animals (e.g., *cat, dog, giraffe, elephant*), six domestic objects (e.g., *basket, nail, corridor*), six environmental elements (e.g., *cloud, liquid, poison*), and six abstract concepts (e.g., *mathematics, peace, symbol*). We publish this association dataset alongside demographic variables including age, gender, profession, country of origin, country of residence, and language fluency.

# 2. MATERIAL AND METHOD

## 2.1. STUDY DESIGN

We collected free associations in an online questionnaire, using the LimeSurvey platform (LimeSurvey Project Team, 2020). Each participant saw all 62 stimulus words (see Stimuli) in one of the seven languages (English, Estonian, French, German, Italian, Lithuanian, and Spanish). Participants were instructed to note the first three words that came to their mind when they thought of this stimulus word. Most participants produced individual words, but some also produced short phrases. Some participants produced fewer than three responses and others more. Overall, the dataset contains within-subject qualitative data (i.e., word responses), which we cleaned and normalised (see Data Preparation). Thus, they are ready for further use.

## 2.2. TIME OF DATA COLLECTION

The data were collected in the following time intervals: i) 30th October 2019 – 8th July 2020; ii) 12th November 2020 – 20th October 2021, and iii) 17th January 2023 – 2nd March 2023.

## 2.3. LOCATIONS OF DATA COLLECTION

The formal data collection was coordinated at the University of Lausanne, Switzerland by DE, DJ, and CM. Participant recruitment from different countries and places were realised by the co-authors in 14 different countries:

- Algeria, Algiers (MB)
- Australia, Adelaide (BB and TL)
- Austria, Graz (CPS) and Vienna (DJ and RYÖ)
- Estonia, Tallinn (MU)
- France, Besançon (EL)
- Germany, Mainz (DO)
- Italy, Padua (MR and CT)
- Lithuania, Kaunas (DJ)
- Mexico, Guadalajara (VC)
- Nigeria, Nsukka (PM)
- Spain, Logroño (EFP)
- Switzerland, Lausanne (DE, DJ, and CM) and Zurich (BL)
- UK, Leeds (JH)
- USA, Boise (MQ)

## 2.4. SAMPLING, SAMPLE, AND DATA COLLECTION

We recruited 1,439 participants – 1,141 women, 290 men, eight participants did not report their gender. We recruited participants of a wide age range resulting in a mean age of 31.47 years ($SD = 13.76$ years, range = 16–80 years). With data extraction, we ensured that participants were native or highly fluent speakers of the language they had completed the survey in. We obtained data in seven languages: English, Estonian, French, German, Italian, Lithuanian, and Spanish (see Table 1 for demographic data per language).

Participants came from different countries, with the most common countries being Switzerland ($n = 235$), Lithuania ($n = 178$), Germany ($n = 152$), Estonia ($n = 122$), Austria ($n = 109$), Mexico ($n = 87$), Spain ($n = 79$), Australia ($n = 63$), Italy ($n = 58$), USA ($n = 58$), Algeria ($n = 57$), UK ($n = 51$), France ($n = 50$), and Nigeria ($n = 47$). The remaining 93 participants came from other countries or indicated two countries (e.g., France and Switzerland) as countries of origin. Most younger participants were university students, while the remaining participants had diverse occupations, all of which appear in the dataset. Participation was voluntary. Participants did not receive monetary remuneration. Some student participants in Switzerland, Germany, and Austria received course credits for taking part in the study.

## 2.5. MATERIAL

### 2.5.1. Stimuli

We selected 62 words as stimuli, falling into the four following domains: i) 11 basic colour terms (Table 2), ii) 5 non-basic colour terms (Tables 2 and 3), iii) 20 emotion terms (Table 2), and iv) 26 'filler' words (i.e., animal names, domestic items, environmental elements, abstract concepts; Table 2).

We took the basic colour terms from colour naming studies (Del Viva et al., 2023; Forbes, 2006; Lindsey & Brown, 2014; Uusküla & Bimler, 2016; Vejdemo et al., 2015) and the 20 emotion terms from the Geneva Emotion Wheel (Scherer, 2005; Scherer et al., 2013). Translations for the basic colour and emotion terms were adopted from Jonauskaite et al. (2020). The five non-basic colour terms were chosen individually for each language based on previous studies (see Lindsey & Brown, 2014; Morgan, 1993; Mylonas & MacDonald, 2016; Uusküla & Bimler, 2020). Two words (i.e., *turquoise*

| SURVEY LANGUAGE | COUNTRY OF ORIGIN | COUNTRY OF RESIDENCE | N | | | | AGE (YEARS) | | |
|---|---|---|---|---|---|---|---|---|---|
| | | | TOTAL | MEN | WOMEN | OTHER | *MEAN* | *SD* | RANGE |
| English | Australia, UK, USA, Nigeria, and other countries | Australia, UK, USA, Nigeria, and other countries | 241 | 59 | 180 | 2 | 37.78 | 15.17 | 18–80 |
| Estonian | Estonia | Estonia and other countries | 123 | 10 | 113 | 0 | 43.75 | 14.15 | 20–74 |
| French | Switzerland, France, Algeria, Portugal, and other countries | Switzerland, France, Algeria, and other countries | 316 | 49 | 266 | 1 | 23.87 | 9.50 | 18–74 |
| German | Germany, Austria, Switzerland, and other countries | Austria, Germany, Switzerland, and other countries | 357 | 69 | 285 | 3 | 25.59 | 8.64 | 18–62 |
| Italian | Italy and other countries | Italy and other countries | 56 | 14 | 42 | 0 | 32.35 | 11.40 | 19–57 |
| Lithuanian | Lithuania and other countries | Lithuania and other countries | 179 | 29 | 150 | 0 | 41.73 | 13.70 | 16–69 |
| Spanish | Mexico, Spain, and other countries | Mexico, Spain, and other countries | 167 | 60 | 105 | 2 | 31.97 | 12.81 | 18–71 |

**Table 1** Demographic information of the studied participants.

*Note.* Countries of origin and of residence are ordered by decreasing sample size (from biggest to smallest). In both cases, we explicitly named countries that had at least 20 participants. 'Other countries' constituted fewer than 20 participants per country in the respective survey language.

| WORD TYPE | ENGLISH | ESTONIAN | FRENCH | GERMAN | ITALIAN | LITHUANIAN | SPANISH |
|---|---|---|---|---|---|---|---|
| Basic colour term | Red | Punane | Rouge | Rot | Rosso | Raudona | Rojo |
| Basic colour term | Orange | Oranž | Orange | Orange | Arancione | Oranžinė | Naranja |
| Basic colour term | Yellow | Kollane | Jaune | Gelb | Giallo | Geltona | Amarillo |
| Basic colour term | Green | Roheline | Vert | Grün | Verde | Žalia | Verde |
| Basic colour term | Blue | Sinine | Bleu | Blau | Blu | Mėlyna | Azul |
| Basic colour term | Purple | Lilla | Violet | Lila | Viola | Violetinė | Violeta |
| Basic colour term | Pink | Roosa | Rose | Rosa | Rosa | Rožinė | Rosado |
| Basic colour term | Brown | Pruun | Brun | Braun | Marrone | Ruda | Marrón |
| Basic colour term | White | Valge | Blanc | Weiß | Bianco | Balta | Blanco |
| Basic colour term | Grey | Hall | Gris | Grau | Grigio | Pilka | Gris |
| Basic colour term | Black | Must | Noir | Schwarz | Nero | Juoda | Negro |
| Non-basic colour term | Turquoise | Türkiissinine | Turquoise | Türkis | Turchese | Turkio spalva | Turquesa |
| Non-basic colour term | Beige | Beež | Beige | Beige | Beige | Smėlinė | Beige |
| Emotion term | Interest | Huvi | Intérêt | Interesse | Interesse | Susidomėjimas | Interesar |
| Emotion term | Amusement | Lõbu | Amusement | Belustigung | Divertimento | Linksmumas | Diversión |
| Emotion term | Pride | Uhkus | Fierté | Stolz | Fierezza | Išdidumas | Orgullo |
| Emotion term | Joy | Rõõm | Joie | Freude | Gioia | Džiaugsmas | Alegría |
| Emotion term | Contentment | Rahulolu | Contentement | Zufriedenheit | Soddisfazione | Pasitenkinimas | Contento |
| Emotion term | Admiration | Imetlus | Admiration | Bewunderung | Ammirazione | Žavėjimasis | Admiración |
| Emotion term | Love | Armastus | Amour | Liebe | Amore | Meilė | Amor |
| Emotion term | Relief | Kergendus | Soulagement | Erleichterung | Sollievo | Nusiraminimas | Alivio |
| Emotion term | Compassion | Kaastunne | Compassion | Mitgefühl | Compassione | Užuojauta | Compasión |
| Emotion term | Pleasure | Nauding | Plaisir | Vergnügung | Piacere | Malonumas | Placer |
| Emotion term | Sadness | Kurbus | Tristesse | Trauer | Tristezza | Liūdesys | Tristeza |
| Emotion term | Guilt | Süü | Culpabilité | Schuld | Colpevolezza | Kaltė | Culpa |
| Emotion term | Regret | Kahetsus | Regret | Bereuen | Rimpianto | Apgailestavimas | Arrepetimiento |
| Emotion term | Shame | Häbi | Honte | Scham | Vergogna | Gėda | Vergüenza |
| Emotion term | Disappointment | Pettumus | Déception | Enttäuschung | Delusione | Nusivylimas | Decepción |
| Emotion term | Fear | Hirm | Peur | Angst | Paura | Baimė | Miedo |
| Emotion term | Disgust | Vastikus | Dégoût | Ekel | Disgusto | Pasibjaurėjimas | Asco |
| Emotion term | Contempt | Põlgus | Mépris | Verachtung | Disprezzo | Panieka | Desprecio |
| Emotion term | Hate | Vihkamine | Haine | Hass | Odio | Neapykanta | Odio |
| Emotion term | Anger | Viha | Colère | Wut | Rabbia | Pyktis | Ira |
| Other | Basket | Korv | Panier | Korb | Cesto | Krepšys | Cesta |
| Other | Cheese | Juust | Fromage | Käse | Formaggio | Sūris | Queso |
| Other | Cloud | Pilv | Nuage | Wolke | Nuvola | Debesis | Nube |
| Other | Liquid | Vedelik | Liquide | Flüssigkeit | Liquido | Skystis | Líquido |
| Other | Nail | Küüs | Ongle | Nagel | Unghia | Nagas | Uña |
| Other | Cat | Kass | Chat | Katze | Gatto | Katė | Gato |
| Other | Dog | Koer | Chien | Hund | Cane | Šuo | Perro |
| Other | Horse | Hobune | Cheval | Pferd | Cavallo | Arklys | Caballo |

(Contd.)

| WORD TYPE | ENGLISH | ESTONIAN | FRENCH | GERMAN | ITALIAN | LITHUANIAN | SPANISH |
|---|---|---|---|---|---|---|---|
| Other | Domestic | Kodune | Domestique | Häuslich | Domestico | Naminis | Doméstico |
| Other | Hood | Kapuuts | Capuche | Kaputze | Cappuccio | Kapišonas | Capucha |
| Other | Routine | Rutiin | Routine | Routine | Routine | Rutina | Rutina |
| Other | Symbol | Sümbol | Symbole | Symbol | Simbolo | Simbolis | Símbolo |
| Other | Corridor | Koridor | Couloir | Korridor | Corridoio | Koridorius | Pasillo |
| Other | Peace | Rahu | Paix | Frieden | Pace | Taika | Paz |
| Other | Ladder | Redel | Echelle | Leiter | Scala | Kopėčios | Escalera |
| Other | Elephant | Elevant | Eléphant | Elefant | Elefante | Dramblys | Elefante |
| Other | Dizzy | Uimane | Etourdi | Schwindelig | Stordito | Apsvaigęs | Mareado |
| Other | Poison | Mürk | Poison | Gift | Veleno | Nuodai | Veneno |
| Other | Hay | Hein | Foin | Heu | Fieno | Šienas | Heno |
| Other | Mathematics | Matemaatika | Mathématiques | Mathematik | Matematica | Matematika | Matemáticas |
| Other | Giraffe | Kaelkirjak | Girafe | Giraffe | Giraffa | Žirafa | Jirafa |
| Other | Squirrel | Orav | Écureuil | Eichhörnchen | Scoiattolo | Voverė | Ardilla |
| Other | Echo | Kaja | Écho | Echo | Eco | Aidas | Eco |
| Other | Bean | Uba | Haricot | Bohne | Fagiolo | Pupelė | Alubia |
| Other | Mouse | Hiir | Souris | Maus | Topo | Pelė | Ratón |
| Other | Tiger | Tiiger | Tigre | Tiger | Tigre | Tigras | Tigre |

**Table 2** Stimulus words in seven languages – direct translations across the studied languages.

| ENGLISH | ESTONIAN | FRENCH | GERMAN | ITALIAN | LITHUANIAN | SPANISH |
|---|---|---|---|---|---|---|
| Lilac | Violetne | Lilas | Violett | Lilla | Alyvinė | Lila |
| Violet | Purpur | Pourpre | Purpur | Porpora | Purpurinė | Púrpura |
| Maroon | Vesihall | Marron | Ocker | Azzuro | Žydra | Rosa |
|  |  |  | Geld |  |  |  |

**Table 3** Remaining stimulus words (mostly non-basic colour terms), in seven languages – not direct translations across the studied languages. See explanations in text for the inclusion of *Geld* as a stimulus word in German.

and *beige*) were translated across the seven languages while three non-basic words, listed in Table 3 (four in German), could vary across the languages. The 'filler' words were taken from Fitzpatrick and colleagues (2015), which two bilingual speakers translated following the back-translation procedure.

During the data collection, we realised that the German word for *yellow* was displayed incorrectly in our survey – *Geld* (meaning 'money') instead of *Gelb* (meaning 'yellow'). We corrected the error and collected an additional sample of German speakers. This is why we have German data on 63 stimulus words (both *Geld* and *Gelb*), but the number of participants responding to *Geld* and *Gelb* is smaller than to the remaining words. Put differently, all participants responded to the same 61 words. Additionally, some participants responded to *Geld* and others to *Gelb*.

### 2.5.2. Procedure

Data collection was conducted by co-authors, who widely distributed the link to the online survey that opened directly in their target language (i.e., the national language of their country). The PDF versions of the surveys in all seven languages are accessible in our repository (see Object and File Names).

The survey started by stating its main goal and providing ethical information (i.e., participation was anonymous and confidential; responses were to be used for research purposes only; participants could stop the survey at any time). Participants provided their consent by clicking on the 'Next' button. After collecting some demographic information (age, gender, profession, country of origin, country of residence, native language and fluency in the language of the survey), participants read the following instructions:

                                                                    

On the screen, you will see one word after the other. For each word, please write down the first three words that come to your mind. For example, you see the word SUN. Then, SKY, YELLOW, and BEAUTIFUL might be the first words that come to your mind. In that case, you would write these words into the word field. There are no right or wrong answers, we are interested in your personal opinion.

Then, participants clicked on the 'yes' button to confirm they understood the task and were ready to continue. In the main part of the survey, they sequentially saw the 62 stimulus words, presented in a semi-randomised order, meaning the three words for *purple*, *violet* and *lilac* never followed each other (due to an a priori research interest, see Epicoco et al., 2021, 2024). For each stimulus word, participants were asked to write down three responses that spontaneously came to their minds when thinking about that word (i.e., three associations). Requesting three associations per stimulus word was intended to elicit a more diverse and heterogeneous set of responses (also see, De Deyne et al., 2013). Participants wrote down the responses in a single response box, with some providing fewer and others providing more than three responses. Everyone provided at least one response per stimulus word. Missing responses were coded as 'NA' in the datasheets. Participants were then thanked and debriefed at the end. They were also given contact details in case they wished to get in touch. The study took around 22 minutes to complete.

## 2.6. QUALITY CONTROL

We implemented several quality control measures throughout the data collection and processing phases to ensure the reliability and validity of our dataset. These measures aimed to produce a robust and reliable free association dataset for cross-linguistic research, with careful attention paid to maintaining both language-specific authenticity and cross-linguistic comparability.

### 2.6.1. Participant Screening and Selection

We asked participants to complete the survey in the language they were highly proficient in. We verified language proficiency by asking all participants to self-report their language fluency on an 8-point scale (1 = not at all, 8 = completely fluent). We excluded participants who reported fluency scores of 6 or lower in the language they completed the survey in. Thus, we arrived at the mean language fluency score of 7.97 (*SD* = 0.18).[1] We also excluded participants, whose responses did not match the language of the survey (e.g., most responses given in English, when the language of the survey is Lithuanian). If participants produced just a couple of responses in a different language (e.g., *touché* in English),

we kept those participants. Finally, during the cleaning process, we ensured that all participants with non-sensical responses were excluded too (e.g., typing *xxx*, *sss*, *ddd*, etc.). Together, we had excluded 98 participants, whose data can be found in the raw data file (see Object and File Names).

### 2.6.2. Survey Design and Implementation

The survey design incorporated several features to ensure consistent data collection across languages. We used a standardised LimeSurvey template (LimeSurvey Project Team, 2020) that was identical across all seven languages, with only the language of presentation differing. Instructions were carefully translated by native speakers and verified by the project coordinators to ensure conceptual equivalence across the surveys. For the presentation of stimulus words, we maintained a semi-randomised order across all languages to control for order effects, particularly ensuring that semantically related colour terms (*purple, violet, lilac*) never appeared consecutively. The survey platform required participants to provide at least one association for each stimulus word before proceeding, thereby ensuring no stimulus word was left entirely without a response.

### 2.6.3. Data Cleaning and Normalisation

Extracting the raw data from the LimeSurvey, we noticed that participant responses for each stimulus word were placed in a single cell. Since most participants provided three responses per stimulus word, we had to separate them. Unfortunately, participants used different ways to indicate the separation between the responses: commas, semicolons, dashes, or simply spaces. Thus, we wrote a custom R code to automatically separate participant responses into different lines (see Object and File Names). Now, each word appeared in a new line, but it meant that multiple word responses (e.g., *summer holidays*) or longer phrases (e.g., *La vie en rose*) were separated too.

Therefore, in the subsequent manual validation step, highly fluent speakers of each language were tasked to put the separated words back into phrases. Afterwards, they cleaned and normalised the responses in the following way:

– **Typo correction.** We corrected words that were spelt incorrectly, unless we could not determine the intended word. In those cases, we left the responses as produced by the participants.
– **Preference for lowercase letters.** We ensured that all the responses were written in lowercase letters unless an uppercase letter was required by the orthography rules of that language (e.g., proper names in English, all nouns in German, etc.).

– **Normalisation of spelling.** In English, we converted American English responses to British English (*gray* became *grey*). We also corrected regional spelling to the national spelling. The latter was most pertinent in German, and we used Duden (https://www. duden.de) as the authority. Words that had multiple accepted spellings (e.g., *meow*, *miaow*, and *meaw* in English) were written in the most common word form.

– **Preference for singular word forms.** When appropriate, we normalised the noun and adjective responses to appear in a singular form (e.g., *cherries* became *cherry*). However, in some cases it was not appropriate because the plural word form was more frequent or the only one correct. In those cases, we kept all responses in plural (e.g., *eyes*, *trousers*, *shoes*).

– **Preference for nouns with no articles.** When appropriate, we deleted articles for nouns. Thus, in English, *the apple* became *apple,* in French, *l'histoire* became *histoire*. We did not delete possessive pronouns (e.g., *my dog*).

– **Preference for nouns in nominative grammatical case.** When appropriate, we normalised noun responses by choosing the nominative case. For instance, in Lithuanian, *atostogoms* became *atostogos*; in German, *des Kindes* became *Kind*; in English, *for peace* became *peace*.

– **Normalisation of adjectives.** Adjectives that occurred without nouns were normalised to appear in the ground form, which sometimes coincided with the masculine form. For instance, in French, *petite* became *petit*, in German, *schnelles* became *schnell*, in Lithuanian, *graži* became *gražu*, etc. Adjectives that occurred with nouns remained accorded to them (e.g., *salle vide*, *weiße Fahne*, *maža mergaitė*).

– **Normalisation of verbs.** Verbs that occurred alone were normalised to appear in the infinitive form. For instance, in German *schlafe* became *schlafen*, in Lithuanian, *važiuoju* became *važiuoti*.

Eleven fluent speakers were involved in total in this task, with 1–4 speakers per language. The entire process was supervised by the project coordinators (DE and DJ), ensuring homogeneity across the languages. When uncertain, speakers discussed the specific instances with each other and/or the project coordinators, favouring standard dictionary spelling and grammar rules.

### 2.6.4. Documentation and Transparency
To facilitate quality assessment by future users, we share all versions of the data (raw, partially processed, and fully cleaned). We created comprehensive codebooks with clear explanations of all variables and processing steps (see FAIR Data and Codebook). We documented all exclusion criteria and their application to ensure reproducibility.

## 2.7. DATA ANONYMISATION AND ETHICAL ISSUES
The study was conducted in accordance with the ethical guidelines (World Medical Association, 2013) and received approval from the ethics committee at the Faculty of Social and Political Sciences, University of Lausanne (C_SSP_032020_00003). All data were collected anonymously, with participants providing informed consent at the beginning of the online survey before proceeding. Participant identifiers were sequentially assigned by LimeSurvey (e.g., first participant assigned ID 1, second ID 2, etc.), ensuring no personally identifiable information was retained. The survey introduction clearly informed participants that their participation was voluntary, anonymous, and confidential, and that their responses would be used for research purposes only. Participants were also informed they could stop the study at any time. The datasets contain no sensitive personal information, with demographic data limited to age, gender, profession, country details, and language information, none of which can be used to identify individual participants.

## 2.8. EXISTING USE OF DATA
Small parts of the dataset have been published or submitted for publication, namely associations with i) three colour terms in French (Epicoco et al., 2021, 2024), ii) 12 colour terms in English (Jonauskaite et al., in revision), and iii) three colour terms in six languages (Epicoco, et al., in prep.). We also proposed a coding system for free associations in Epicoco et al. (2021, 2024), inspired by the works of Rosch (1973) and Griber et al. (2018).

## 3. DATASET DESCRIPTION AND ACCESS

### 3.1. GENERAL OVERVIEW OF THE DATASET
Our participants produced 223,786 responses in total. On average, there are 31,969 responses per language and 516 responses per stimulus word (see Table 4). We also determined the number of discrete responses per language and per stimulus word. By discrete responses, we refer to an 'idea', thus, ignoring how often an idea was produced. For instance, if the response COLOUR was produced 100 times in a language-specific dataset, COLOUR constituted a single discrete response (see Table 4). We report discrete responses for each language-specific dataset. To have a general overview of the obtained responses, in Figure 1, we display all the discrete responses in the form of word clouds, separately for each language. Larger font sizes indicate more frequent discrete responses.

Jonauskaite et al. *Journal of Open Psychology Data* DOI: 10.5334/jopd.140

| SURVEY LANGUAGE | PARTICIPANTS | RESPONSES | | DISCRETE RESPONSES | |
|---|---|---|---|---|---|
| | | TOTAL | PER STIMULUS WORD | TOTAL | PER STIMULUS WORD |
| English | 241 | 33,415 | 538.95 | 10,757 | 173.50 |
| Estonian | 123 | 17,946 | 289.45 | 8,242 | 132.94 |
| French | 316 | 51,441 | 829.69 | 12,501 | 201.63 |
| German | 357 | 61,528 | 992.39 | 15,417 | 248.66 |
| Italian | 56 | 9,190 | 148.23 | 4,773 | 76.98 |
| Lithuanian | 179 | 24,008 | 387.23 | 9,754 | 157.32 |
| Spanish | 167 | 26,258 | 423.52 | 8,924 | 143.94 |
| All languages | 1,439 | 223,786 | 3,609.45 | 70,368 | 1,134.97 |

**Table 4** The number of participants, total responses and total discrete responses per language, across all stimulus words (total) and on average per stimulus word.

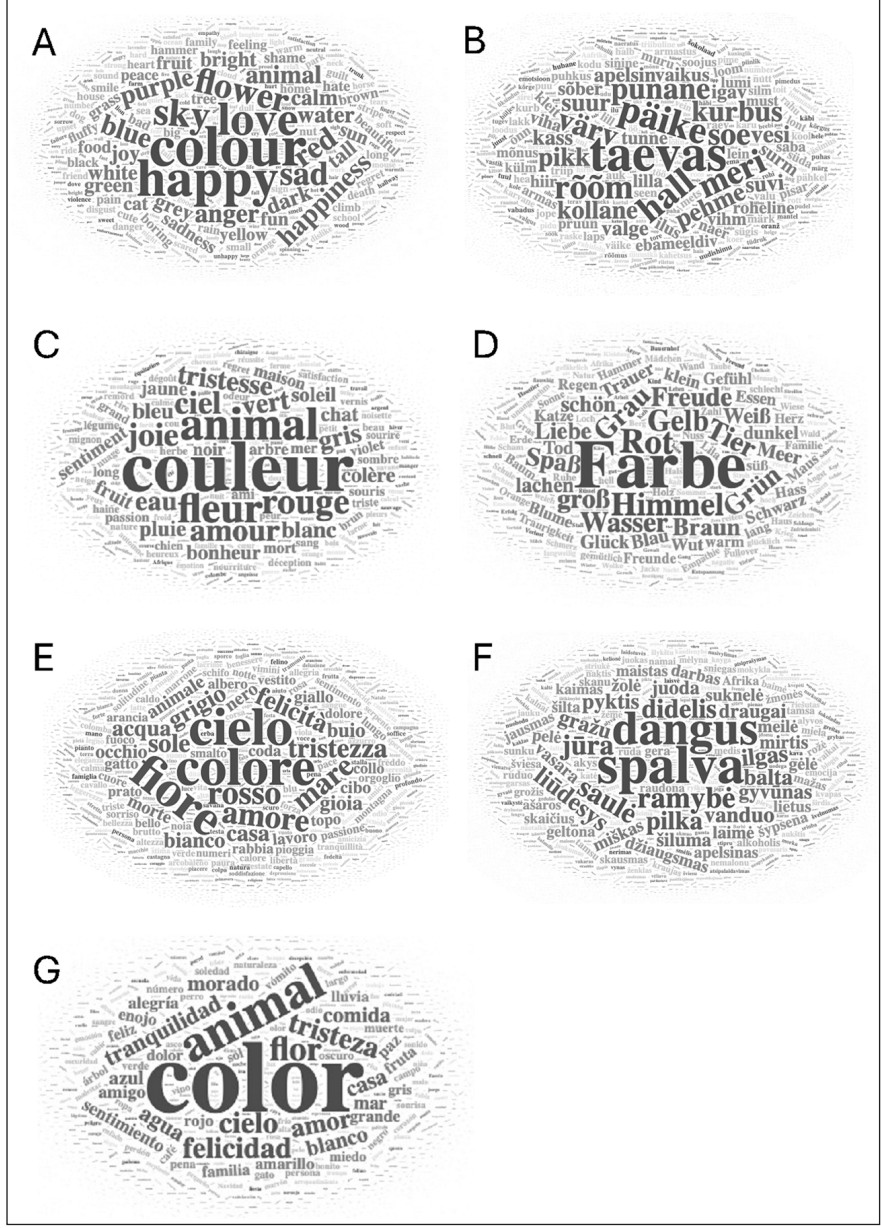

**Figure 1** Word clouds in English **(A)**, Estonian **(B)**, French **(C)**, German **(D)**, Italian **(E)**, Lithuanian **(F)**, and Spanish **(G)**. Larger words indicate more frequent responses in the overall dataset of each language.

## 3.2. REPOSITORY LOCATION

The research material and the data are accessible on OSF: https://osf.io/xzcbg/. The project is identified with the DOI: https://doi.org/10.17605/OSF.IO/XZCBG.

## 3.3. OBJECT AND FILE NAMES

The research material and the datasets are classified in the following folders:

– **Datasets**
  – **Cleaned data in seven languages**: Contains cleaned and normalised data in seven languages (one file per language, plus one file with all languages, where stimulus words are direct translations). Each participant's data span several rows. Each cleaned participant response appears on a separate row, and demographic data are repeated across the rows. The column 'Association number' indicates whether this was their first, second, third, etc. response. Compared to the raw and uncleaned datasets, we recoded language and country codes using the international standardisation codes. *This dataset is recommended for analyses*.
  – **Raw data – all languages together**: Contains a single file with all the raw data. The file was downloaded directly from LimeSurvey and was not modified. Each participant's data occupy a single row. The file includes data from all participants, including those excluded from the final dataset during the cleaning procedure (see Sampling, Sample, and Data Collection). This dataset can be matched with the other datasets using participant ID (Response ID -> PS_ID).
  – **Uncleaned data in seven languages**: Contains uncleaned data in seven languages (one file per language except for German, which has two files – see the explanation under Surveys). Each participant's data span multiple rows, and demographic data are repeated. Each word a participant produced appears in a new row, in the order they had been typed. Words were extracted automatically, meaning that responses with more than one word appear on separate rows (e.g., *light blue* would be separated into two rows with *light* and *blue* as responses). Such errors were corrected during the cleaning process. This dataset can be matched with the other datasets using participant ID (Response ID -> PS_ID).
– **Stimuli and Codebook**
  – **Codebook_cleaned_dataset_variables.csv**: Contains variable names and their descriptions for the cleaned datasets.
  – **Codebook_datasets.csv**: Contains dataset organisation information, to help readers navigate the files (repeats some information from above).
  – **Stimulus_word_translations.csv**: Stimulus words in all seven languages (identical to those in Tables 2 and 3).
  – **Convert raw to uncleaned.R:** a custom R code, converting the raw data to the uncleaned data.
  – **Surveys**: Contains LimeSurvey PDFs in seven languages. There are two files for German: German_1 contains *Geld* ('money'), and German_2 contains *Gelb* ('yellow') as stimulus words.

## 3.4. DATA TYPE

The folder 'Surveys' includes research materials – LimeSurvey surveys in all seven languages. The folder 'Raw data – all languages together' includes primary raw and unprocessed data. The folder 'Uncleaned data in seven languages' includes primary data, after the first processing step. The folder 'Cleaned data in seven languages' includes primary processed data, after the data cleaning and normalisation. Generally, we recommend researchers in psychology to use data from the 'Cleaned data in seven languages' folder, unless their research question requires the raw data (e.g., studies examining the typing choices of respondents, typos, exact word forms, etc.).

## 3.5. FORMAT NAMES AND VERSIONS

The folder 'Surveys' includes research materials in PDF format, that can be opened with standard document readers (e.g., Adobe Acrobat Reader, Preview, etc.). The folders under 'Datasets' include unprocessed and processed datasets in the XLSX and CSV formats. The XLSX files can be opened with Microsoft Excel or free alternatives, such as LibreOffice Calc, OpenOffice Calc, or Google Sheets, while the CSV files can additionally be opened with any text editor. Both, XLSX and CSV files can also be directly imported into R, Python, Matlab, SPSS, Stata, etc. The stimulus words in all seven languages (see Stimuli) and the relevant codebooks (see FAIR Data and Codebook) appear in 'Stimuli and Codebook' folder, in CSV format.

## 3.6. LANGUAGE

Variable names that are common across the languages (e.g., participant ID, country of residence, country of origin, etc.) are displayed in British English. The stimulus words are displayed in the language of the survey – English, Estonian, French, German, Italian, Lithuanian, or Spanish. Participant responses are also displayed in the language of the survey.

## 3.7. LICENCE

The data are published under the CC-BY 4.0 license.

### 3.8. LIMITS TO SHARING

There is no identifying information in the dataset. All participant IDs were generated by LimeSurvey in the order that participants took the survey. Therefore, there are no barriers to the full sharing of the dataset, which is fully accessible with no restrictions. It is not under embargo.

### 3.9. PUBLICATION DATE

The data were published in the repository on 2nd June 2025.

### 3.10. FAIR DATA AND CODEBOOK

Our dataset adheres to the FAIR (Findability, Accessibility, Interoperability, and Reuse) principles in the following way (Wilkinson et al., 2016). For Findability, we have assigned a persistent DOI through the OSF repository, mentioned in the Repository location section. Accessibility is ensured through the open access, described in the Limits to Sharing section, and the choice of the data repository (OSF), which guarantees data preservation and long-term access. Interoperability is supported through the XLSX format, which is machine-readable and compatible with diverse software. Reusability is facilitated through the data cleaning and the file documentation, outlined in the Data Cleaning and Normalisation and Object and File Names sections respectively.

The dataset codebook, available as 'Codebook_ cleaned_datasets.xlsx' in the repository, supplements the file structure information provided in the Object and File Names section. It contains complete descriptions of all variables including participant demographics (as referenced in the Sampling section), stimulus information (detailed in the Stimuli section), and response data (described in the General overview of the dataset section). The codebook also provides specific notes on language-specific features that complement the information provided in the Language section. The codebook 'Codebook_datasets.xlsx' facilitates navigation of the different datasets.

## 4. REUSE POTENTIAL

This dataset has substantial reuse potential across multiple disciplines and research contexts. It is a cross-linguistic dataset of free associations from seven languages. Its detailed documentation and standardised methodology make it an accessible and valuable resource for researchers investigating the intersection of language, thought, and culture. The primary strength lies in its comprehensive nature: it contains responses from 1,439 participants across 14 countries to 62 stimulus words, yielding a rich cross-linguistic repository of semantic associations. It can help investigate diverse research questions in various disciplines. Thus, our

suggestions below are indicative, but not exhaustive. The data have been cleaned and normalised, making them ready for analysis.

### 4.1. COGNITIVE PSYCHOLOGY AND PSYCHOLINGUISTICS

This dataset is useful in the study of semantic networks and meaning construction within and across languages and cultures. It is particularly useful for those interested in colour and emotion terms (e.g., see colour semantics in Schloss, 2024), and the interplay between the two (e.g., see Jonauskaite & Mohr, 2025; Ram et al., 2020). The dataset includes associations for basic and non-basic colour terms enabling a better understanding of how colour knowledge is organised (e.g., Epicoco et al., 2024; Mervis & Roth, 1981; Mylonas & Griffin, 2020). One can also compare associations for the same stimulus word across dialects or different language communities (e.g., Epicoco et al., 2021, 2024). Perhaps, such studies might help understanding historical and post-colonial influences (Levisen & Sippola, 2019; Wassmann, 2017) or contribute to anthropological research examining cultural values and meaning systems (Handwerker, 2002). This dataset supports the decolonisation of research practices in psychology by providing a valuable cross-linguistic resource, enabling researchers to move beyond an Anglocentric approach. Then, researchers can study and compare semantic universals and language-specific conceptualisations, like those relevant to idiomatic and metaphoric expressions (e.g., Oguz, 2025). Yet, analysis can go beyond colour and emotion terms, exploring the semantic structure of other natural and human-made categories that are part of the stimulus word set.

The cleaned and the raw datasets allow the analysis of response patterns. Using the cleaned data, researchers can tap into divergent thinking and creativity, for which spontaneous cognition and associative flexibility are central (Beaty & Kenett, 2023; Rossmann & Fink, 2010). Free association data can be analysed for fluency, originality, semantic distance and other features (e.g., Abu-Akel et al., 2020; Beaty et al., 2022; Merseal et al., 2025). The raw dataset shows variations in i) how words were typed (lowercase, capital letters, errors), ii) how language was used (e.g. dialect, foreign terms), and iii) other intra- and inter-individual variations, all of which yield information on behavioural thought patterns. Researchers could further reduce the complexity of the datasets by compiling participants' responses into smaller entities. Epicoco et al. (2024) proposed a coding scheme of nine semantic themes, inspired by the earlier studies (Paramei et al., 2018; Rosch, 1973). Showing the potential of this approach, we previously applied this coding scheme to analyse i) the French colour terms *violet*, *lilas*, and *pourpre* in three countries (Epicoco et al., 2021, 2024) and ii) the reasons for colour preferences (Epicoco, et al., in revision).

## 4.2. CLINICAL AND DEVELOPMENTAL PSYCHOLOGY

This dataset could also serve as normative data for clinical research investigating semantic processing in various clinical populations such as schizophrenia, autism spectrum disorders, or depression (e.g., Pintos et al., 2022). For developmental research, data from younger age groups could be added (see Cox & Haebig, 2022) to examine how semantic knowledge develops from childhood through adulthood and old age (e.g., Citraro et al., 2023; Fourtassi et al., 2020; Unger & Fisher, 2021), or how associative patterns differ between native and non-native speakers to identify language learning trajectories (e.g., Kan et al., 2024). Shifts in emotional and non-emotional meaning-making could be detected, such as developmental trends in the representation of key affective concepts like *love* or *anger* (Nook et al., 2018). On the other end of the age continuum, researchers could investigate cognitive ageing directly within the current dataset, as we recruited participants from 16 to 80 years old (e.g., Jonauskaite et al., 2024).

## 4.3. TRANSLATION STUDIES

This dataset enables cross-linguistic analysis of semantic associations across Indo-European languages (English, French, German, Italian, Lithuanian, Spanish) and a Finno-Ugric language (Estonian). It can provide insights into translation theory and practice (Baker, 2018). Professional translators could benefit from this resource such as when having to i) translate connotative meanings of notoriously difficult to translate words (e.g., Uusküla, 2020; Wassmann, 2017) or ii) dealing with cognitive processes underlying translation difficulties in idiomatic language (e.g., Kalda & Uusküla, 2019; Oguz & Uusküla, 2023).

## 4.4. EDUCATION

Free association patterns could support the development of teaching materials that reflect how native speakers conceptually organise vocabulary (e.g., Curry & Mark, 2024; Looi et al., 2025). Knowing the most common associations that native speakers make with particular words, language instructors could design vocabulary learning activities that are supported by cognitive organisation rather than arbitrary word lists (e.g., Kismetova & Serikova, 2024). The cross-linguistic nature of the dataset is useful for comparative language education to highlight similarities and differences in how concepts are linked within and across languages (e.g., Ivanova, 2024).

## 4.5. MARKETING

Free associations could help understand the emotional, symbolic, and otherwise abstract meaning of products, brands, and services in consumer contexts (Rahman & Areni, 2016; Vriens et al., 2019). Analysing the affective and sensory associations of personal experiences and representations might help making advertisements more memorable and personally relevant (see also Sester et al., 2013). The dataset incorporates different demographic variables and encompasses participants from a wide age range. Hence, the current data might help tailor messages to selected audiences and causes (Olsen & Pracejus, 2020; Yang & Li, 2016).

## 4.6. LIMITATIONS AND CONSIDERATIONS FOR REUSE

Potential users should consider several limitations when reusing this data. While the sample size is substantial ($N = 1{,}439$), the distribution across languages is uneven, with larger samples for i) French and German, and ii) women than men. The sample sizes across the different countries are also uneven. Then, French and German samples had more young university students than samples in the other languages, potentially limiting age and socio-economic diversity (of which we have no information). Finally, data collection occurred between 2019 and 2023, which could introduce temporal effects, as semantic associations are constantly evolving (e.g., Unger & Fisher, 2019).

## 4.7. FUTURE POTENTIAL

This dataset can serve as a foundation for future projects. It could be expanded to include additional languages, used in longitudinal studies that track changes in semantic associations over time, combined with neuroimaging studies to investigate the neural correlates of cross-linguistic semantic processing, and integrated with other cross-cultural datasets to examine broader patterns of cognition. The dataset could also be used as training and testing data for computational linguistics or to compare with the free associations generated by large language models (see Abramski et al., 2025). It could support the development and evaluation of semantic models across multiple languages, improvement of machine translation systems by incorporating culturally specific semantic associations, enhancement of natural language understanding in multilingual contexts, and testing of computational models of semantic networks. The cleaned, normalised format makes this dataset immediately usable for computational applications without extensive pre-processing. By providing well-constructed, cross-linguistic datasets, it encourages decolonisation in the research practice.

## NOTE

1 Our subjective measure of language fluency gives some indication of how well participants understood the language of the survey. Yet, self-report measures are prone to biases (e.g., Hernández-Rivera et al., 2024). Thus, this score should be interpreted with caution.

## ACKNOWLEDGEMENTS

We thank Jaanika Aru, Allison Creed, Giancarlo Epicoco, Eugénie Gollut, Christina Ila, Aurika Jonauskienė, Benjamin Kasslatter, Esther Liekmeier, Nizar Michaud, Yanisha Soborun, and Derya Sönmez for their help with translation or data cleaning.

## FUNDING STATEMENT

This research was possible thanks to the Swiss National Science Foundation, providing career fellowship grants to DJ (P0LAP1_175055; P500PS_202956; P5R5PS_217715; PZ00P1_223781) and a project funding grant to CM (100014_182138), funding DE's doctoral studies. We also thank the ERASMUS+ fellowships to RYÖ and SS, funding their research visits at the University of Vienna, under the supervision of DJ.

## COMPETING INTERESTS

The authors have no competing interests to declare.

## AUTHOR CONTRIBUTIONS

- Conceptualisation: DJ, DE, MU, CM
- Data curation: DJ, DE
- Formal analysis (data cleaning): DJ, DE, VC, CGD, SLC, LM, MO, CPS, PQ, SS, GFMS, MT, MU
- Funding acquisition: DJ, CM
- Investigation (data collection): DJ, DE, MB, BB, VC, EFP, JH, EL, BL, TL, PM, DO, RYÖ, CPS, MQ, MR, CT, MU, CM
- Methodology: DJ, DE, MU, CM
- Project administration: DJ, CM
- Resources: DJ, CM
- Software: DJ, DE
- Supervision: DJ, CM
- Validation: DJ, DE, VC, MU, CM
- Visualization: DJ, DE
- Writing – original draft: DJ, DE, CM
- Writing – review & editing: DJ, DE, MB, BB, VC, EFP, CGD, JH, EL, BL, TL, SLC, LM, PM, DO, MO, RYÖ, CPS, PQ, MQ, MR, SS, GFMS, MT, CT, MU, CM

## AUTHOR AFFILIATIONS

**Domicele Jonauskaite** orcid.org/0000-0002-7513-9766
Institute of Psychology, University of Lausanne, Lausanne, Switzerland; Faculty of Psychology, University of Vienna, Vienna, Austria

**Déborah Epicoco** orcid.org/0000-0001-8683-3000
Institute of Psychology, University of Lausanne, Lausanne, Switzerland; School of Social Work, University of Applied Sciences and Arts of Western Switzerland (HETSL | HES-SO), Lausanne, Switzerland

**Maliha Bouayed Meziane** orcid.org/0000-0003-1813-7702
Preparatory Classes Department, École Nationale Polytechnique, Algiers, Algeria

**Britt Burton** orcid.org/0000-0002-9331-1160
Justice and Society, University of South Australia, Adelaide, Australia

**Violeta Corona** orcid.org/0000-0003-3410-7693
Facultad de Ciencias Económicas y Empresariales, Universidad Panamericana, Zapopan, Mexico

**Eduardo Fonseca-Pedrero** orcid.org/0000-0001-7453-5225
Department of Educational Sciences, University of La Rioja, Logroño, Spain

**Consuelo González-Dávila** orcid.org/0009-0008-2483-1446
Faculty of Psychology, National Autonomous University of Mexico (UNAM), Mexico City, Mexico

**Jelena Havelka** orcid.org/0000-0002-7486-2135
School of Psychology, University of Leeds, Leeds, UK

**Eric Laurent** orcid.org/0000-0002-7112-6765
UMR INSERM 1322 LINC & UAR 3124 CNRS MSHE Ledoux, Université Marie et Louis Pasteur, Besançon, France

**Bigna Lenggenhager** orcid.org/0000-0003-0418-9931
Department of Psychology, University of Zurich, Zurich, Switzerland; Association for Independent Research, Zurich, Switzerland

**Tobias Loetscher** orcid.org/0000-0003-1967-2926
Justice and Society, University of South Australia, Adelaide, Australia

**Stephanie Lopez Castiñeira**
Department of Psychology, University of Konstanz, Konstanz, Germany

**Leila Manni** orcid.org/0009-0003-3717-2797
Institute of Psychology, University of Lausanne, Lausanne, Switzerland

**Philip Mefoh** orcid.org/0000-0002-7529-3843
Department of Psychology, University of Nigeria, Nsukka, Enugu State, Nigeria

**Daniel Oberfeld** orcid.org/0000-0002-6710-3309
Institute of Psychology, Johannes Gutenberg University Mainz, Mainz, Germany

**Merle Oguz** orcid.org/0000-0002-2436-8403
School of Humanities, Tallinn University, Tallinn, Estonia

**Rabia Yağmur Özduran**
Faculty of Psychology, University of Vienna, Vienna, Austria; Faculty of Human and Social Sciences, Yasar University, Izmir, Turkey

**Corinna Perchtold-Stefan** orcid.org/0000-0002-8334-0574
Department of Psychology, University of Graz, Graz, Austria

**Patricia Quant** orcid.org/0009-0005-1927-4380
Faculty of Life Sciences, University of Vienna, Vienna, Austria

**Michael Quiblier**
Institute of Psychology, University of Lausanne, Lausanne, Switzerland

**Maja Roch** orcid.org/0000-0001-5604-9557
Department of Developmental Psychology and Socialization, University of Padova, Padova, Italy

**Sude Sarayköylü** orcid.org/0000-0002-2347-4466
Faculty of Psychology, University of Vienna, Vienna, Austria; Department of Psychology, Carl von Ossietzky University Oldenburg, Oldenburg, Germany

**Giulia F. M. Spagnulo** ⓘ orcid.org/0009-0000-7200-7867

Institute of Psychology, University of Lausanne, Lausanne, Switzerland

**Maël Theubet** ⓘ orcid.org/0009-0001-4650-068X

Institute of Psychology, University of Lausanne, Lausanne, Switzerland

**Cecilia Toscanelli** ⓘ orcid.org/0000-0001-8808-3506

Institute of Psychology, University of Lausanne, Lausanne, Switzerland; Institute of Psychology, University of Bern, Bern, Switzerland; Research Group Work & Organisational Psychology (WOPP-O2L), KU Leuven, Leuven, Belgium

**Mari Uusküla** ⓘ orcid.org/0000-0001-6219-9228

School of Humanities, Tallinn University, Tallinn, Estonia

**Christine Mohr** ⓘ orcid.org/0000-0002-3720-7115

Institute of Psychology, University of Lausanne, Lausanne, Switzerland

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

## PEER REVIEW COMMENTS

*Journal of Open Psychology Data* has blind peer review, which is unblinded upon article acceptance. The editorial history of this article can be downloaded here:

- **PR File 1.** Peer Review History. DOI: https://doi.org/10.5334/jopd.140.pr1

**TO CITE THIS ARTICLE:**
Jonauskaite, D., Epicoco, D., Meziane, M. B., Burton, B., Corona, V., Fonseca-Pedrero, E., González-Dávila, C., Havelka, J., Laurent, E., Lenggenhager, B., Loetscher, T., Castiñeira, S. L., Manni, L., Mefoh, P., Oberfeld, D., Oguz, M., Özduran, R. Y., Perchtold-Stefan, C., Quant, P., Quiblier, M., Roch, M., Sarayköylü, S., Spagnulo, G. F. M., Theubet, M., Toscanelli, C., Uusküla, M., & Mohr, C. (2025). Free Association Database for a 62-Word Dataset Including Emotion and Colour Terms in English, Estonian, French, German, Italian, Lithuanian, and Spanish: Data from 14 Countries. *Journal of Open Psychology Data,* 13: 4, pp. 1–17. DOI: https://doi.org/10.5334/jopd.140

**Submitted:** 02 June 2025      **Accepted:** 18 July 2025      **Published:** 04 August 2025

