## [Peer Review History. · Journal of Open Psychology Data]

Peer Review Comments from "Free Association Database for a 62-Word Dataset Including Emotion and Colour Terms in English, Estonian, French, German, Italian, Lithuanian, and Spanish: Data from 14 Countries"

Dear Domicela Jonauskaite, Déborah Epicoco, Maliha Bouayed Meziane, Britt Burton, Violeta Corona, Eduardo Fonseca-Pedrero, Consuelo González-Dávila, Jelena Havelka, Eric Laurent, Bigna Lenggenhager, Tobias Loetscher, Stephanie Lopez Castiñeira, Leila Manni, Philip Mefoh, Daniel Oberfeld, Merle Oguz, Rabia Yağmur Özduran, Corinna Perchtold-Stefan, Patricia Quant, Michael Quiblier, Maja Roch, Sude Sarayköylü, Giulia F. M. Spagnulo, Maël Theubet, Cecilia Toscanelli, Mari Uusküla, Christine Mohr,

After review, we have reached a decision regarding your submission to Journal of Open Psychology Data, "Free Association Database for a 62-Word Dataset Including Emotion and Colour Terms in English, Estonian, French, German, Italian, Lithuanian, and Spanish: Data from 14 Countries". Our decision is to request revisions of the manuscript prior to acceptance for publication.

In summary, the two reviewers consider the work and data to have significant value, but have proposed some helpful suggestions which could help with the clarity of the manuscript and the accessibility of the dataset. As such, I ask you to consider these suggestions systematically and provide a document outlining your response to each comment and any taken actions. The full review information is included at the bottom of this email. I do hope you find this all constructive and supportive of the ambitions of the work.

Instructions for how to resubmit your article online are pasted below. Please ensure that your revised files adhere to our author guidelines, and that the files are fully proofed prior to upload. Please also include a revised version of your article with 'tracked changes', adding comments where appropriate, to indicate the revisions made, in addition to a brief document outlining how you have responded to the reviewers' requests.

If you have trouble processing the revisions, our Help Center (<https://help.u-community.io>) or downloadable PDF (<https://bit.ly/Author-Guide-OJS-3>) may be able to help. If not, please get in touch and we'll be happy to help.

Please also ensure that all copyright permissions have been attained for any figures/tables you have included.

Please could you have the revisions submitted with four weeks. If you cannot make this deadline, please let us know as early as possible.

Kind regards,

Prof Thomas Rhys Evans

Reviewer B:

Recommendation: Revisions Required

Comments to the author(s)

JOPD Review

Free Association Database for a 62-Word Dataset Including Emotion and Colour Terms in English, Estonian, French, German, Italian, Lithuanian, and Spanish: Data from 14 Countries

Summary

This data paper describes a multilingual dataset of word associations obtained from 14 countries and 7 languages. The dataset has high potential for informing psycholinguistic research on semantic alignment across cultures, an important and timely topic. Overall, the manuscript is very clearly written and the data repository meets the publication standards and requirements of JOPD. Below are a few suggestions and comments for consideration.

Specific Comments on Manuscript

p. 4 - "imaginability" -> "imageability"? "affectivity" -> "valence/affect" My suggestions are based on the terms I tend to see more frequently in the literature.

p. 5 / Section 2.2 - Should the time stamp of data collection for each participant be provided in the cleaned dataset itself? It was also mentioned further below (Section 4.6) about the possibility of semantic evolution leading to changes in associative patterns and the data appears to be collected over a very long period.

p. 6 / Section 2.4 - I am bit confused about this point: "we ensured that participants had completed the survey in their native language or the national language of their country" What happens in situations where the participant's native language is not the national language of their current residence? e.g., an Italian native speaker living in Australia. Did they get to chose to the do the study in Italian or English? From the cleaned datasets I see that sometimes participants completed the study in their non-native language. Perhaps this point can be explained further.

p. 16 / Section 2.6.1 - Recent work has shown that self-reported language proficiency can be subject to biases and hence objective measures are recommended (Hernández-Rivera et al., 2024), hence it may be worth being a bit more cautious in claiming that the self-evaluation selection criteria is strict.

Hernández-Rivera, E., Kalogeris, A., Tiv, M., & Titone, D. (2024). Self-evaluations and the language of the beholder: Objective performance and language solidarity predict L2 and L1 self-evaluations in bilingual adults. *Cognitive Research: Principles and Implications*, 9(1), 75. <https://doi.org/10.1186/s41235-024-00592-4>

Specific Comments on Data Repository

Codebook_cleaned_datasets.xlsx

"Participant country of origin (for country codes, see xxx)" The country codes are missing? Under variable description for PS_ID, may be useful to repeat the information about the mapping of the ID variables across data files here. (i.e., ID in the uncleaned dataset => PS_ID...)

I am confused about the inclusion of the "Semantic coding scheme.pdf" file - since it was not used to code the responses here, as far as I can tell. Could the rationale for including it be made clearer?

Reviewer F:
Recommendation: Revisions Required

Comments to the author(s)

The manuscript describe a data set containing free associations made on the basis of a list of 62 words (+1 in german due to an error) provided by 1,439 participants in 7 different languages (14 different countries). In total, 223,786 free associations are reported (some participants provided less than 3 associations, and some provided more). The data is shared in three versions: raw (direct LimeSurvey output), uncleaned, and cleaned + normalised.

I have no doubt that this data set is highly valuable - and I therefore recommend the publication of the data set. However, I recommend some minor revisions.

The authors provided a thorough description of data collection and data cleaning process. Some aspects could be described more.

- Cleaning and normalisation process: please specify how many fluent speakers took part in the process for each language and how possible conflicts between speakers were solved.
- "The non-basic colour terms were frequent words in each language, but they were not direct translations" - Please elaborate what you mean by "were not direct translations".
- Response Times (or rather Time spent generating words for each prime) would be a welcomed addition in the data set as it might inform to what extent the free-associations were spontaneous (vs deliberated). However, my understanding is that this information was not collected (i.e., absent from the raw limesurvey output). If I am wrong however, it might be a nice addition.
- I have reservations regarding the xlsx format that was chosen: it is a proprietary and opaque format and does not guarantee inter-operability. While .xlsx is indeed widely supported today, it is a proprietary format tied to Microsoft's ecosystem. This raises long-term concerns about sustainability and accessibility, as continued access depends on third-party software compatibility and vendor support. In contrast, open, text-based formats like .csv offer broader interoperability and better guarantees for archival stability. While I do understand that languages including accents might be complicated to encode - I believe it would be possible to use UTF-8 encoding to preserve format in csv or txt. This option could be done in addition to the excel format.
- Furthermore, loading xlsx in R is only made possible using a (non-base) R package

(readxl). This should be specified in the relevant section ("The XLSX files can also be directly imported into R, Python, Matlab, SPSS, Stata, etc."). This further shows that excel is not the most inter-operable format.

- Accessibility also includes long-term access (e.g., repository guarantees, data preservation), which is not mentioned.
- The provided coding scheme (Epicoco et al., 2024) is interesting, and I am not questioning its relevance. However, other coding schemes exist (e.g., LIWC), and including many details about this specific scheme (and uploading an online guide in the repository, 'Semantic Coding Scheme.pdf') risks making the data sharing and manuscript less concise and direct. I would recommend removing the unnecessary documents (Semantic Coding Scheme.pdf) and descriptions (table 5) of this scheme -- it is perfectly acceptable to reference it though.
- Note a typo: "pourpe" should be read as "pourpre".
- Sharing the R scripts used to process the raw data would be a valuable addition, significantly enhancing the transparency and reproducibility of the dataset.
- Codebook of the cleaned data set includes "for country codes, see *xxx*", please update.
- Ideally, reasons for exclusions would be documented in the data file.
- I would recommend adding a master file containing all languages all together.
- I note that the french file contains 4 occurrences of "Comissions" and one occurrence "Commissions (dans le sens des courses)", and in the uncleaned data file, I can see that these corresponds to: comissions (k = 1), commission (k = 1), Commission (k = 1), commissions (k = 2). It is unclear why and how the coders created an isolated 'commission (dans le sens des courses)' (see also in the colour Orange, in French, coding of 'orange' vs 'orange (fruit)'). This might be an isolated oddity. It might be worth double checking each data file to make sure that normalisation is transparently justified - and accurately conducted.

Overall, this is a very interesting and rich data set that undoubtedly deserves to be published. The manuscript is quite clear and I enjoyed reading it. The data has a lot of potential and is quite amusing to explore. I want to congratulate the authors for their work.